# Understanding the Effect of Multiple Sclerosis on General and Dimensions of Mental Health

**DOI:** 10.3390/jcm11247483

**Published:** 2022-12-16

**Authors:** Weixi Kang

**Affiliations:** UK DRI Care Research and Technology Centre, Department of Brain Sciences, Imperial College London, 926, Sir Michael Uren Hub, 86 Wood Lane, London W12 0BZ, UK; weixi20kang@gmail.com

**Keywords:** multiple sclerosis, MS, GHQ-12, social dysfunction and anhedonia, depression and anxiety, loss of confidence

## Abstract

Objective: The objective of the current study is to investigate how general and dimensions of mental health are affected by multiple sclerosis (MS). Methods: Factor analysis, generalized linear models, and one-sample t-tests were used to analyze data from 78 people with MS with a mean age of 52.19 (S.D. = 12.94) years old and 25.64% males and 38,516 people without MS with a mean age of 49.10 (S.D. = 18.24) years old and 44.27% males from Understanding Society. Results: The current study found that there are three underlying factors of the GHQ-12 labeled as GHQ-12A (social dysfunction and anhedonia; 6 items), GHQ-12B (depression and anxiety; 4 items), and GHQ-12C (loss of confidence; 2 items), and the general mental health, GHQ-12A (social dysfunction and anhedonia), and GHQ-12C (loss of confidence) are associated with MS. Conclusions: Effective mental health management in MS patients is important given mental health in people with MS is linked to the onset of MS and exacerbating disease progression/relapses.

## 1. Introduction

Multiple sclerosis (MS) is a severe neurological condition in which the loss of myelin sheaths and subsequent neurodegeneration cause disability in patients over time. People with MS have many visible (e.g., walking difficulties; [1] and internally invisible symptoms [2,3]. Invisible symptoms may include “fatigue, mood and mental health disorders, cognitive changes, pain, bowel/bladder dysfunction, sexual dysfunction, and vision changes” [4]. Understanding invisible symptoms such as mental health is crucial to avoid misunderstandings (e.g., that MS patients only suffer from visible symptoms), stigma, and gaps in care/accommodations [5,6].

Indeed, people with MS are more likely to have certain mental health comorbidities than people without MS, which may include elevated chronic stress, depression, anxiety, psychosis, bipolar disorder, adjustment disorder, and suicidal intentions [7,8]. Stress and depression can increase the challenges of managing bipolar disorders, which are more prevalent in individuals living with MS [9]. Accumulating evidence has suggested that mental health could have an impact on one’s quality of life and disease progression in MS patients [10,11,12]. However, mental health in people with MS may be under-studied and under-treated [13,14].

The underlying mechanisms that could explain the association between MS and mental health are complex and consist of multiple factors, which include neural substrates affected by MS and the challenges of living with MS [4]. For instance, as reviewed by [8], lesion location, endocrine systems, and medication could affect mental health. In addition, the interplay of stigma and other invisible symptoms could result in worse mental health (see [5] for a review).

First developed by Goldberg in the 1970s, the general health questionnaire (GHQ) has been considered as a reliable measure of mental health. Among other versions of the GHQ, the GHQ-12 has been widely used, which includes 12 items, each of which is assessed with a Likert scale [15]. Many studies have evaluated the psychometric property of this questionnaire [16,17,18,19,20,21] and proven that GHQ-12 has good specificity, reliability, and sensitivity [22,23]. Many studies have explored the factor structure of the GHQ-12, although the GHQ-12 was initially developed as a unidimensional scale. Between two- or three-factor models, there is a lot of empirical support behind a three-factor model of the GHQ-12 [24,25,26,27,28], which includes GHQ-12A (social dysfunction and anhedonia; 6 items), GHQ-12B (depression and anxiety; 4 items), and GHQ-12C (loss of confidence; 2 items). A typical argument made that favors the use of the unidimensional model of the GHQ-12 rather than the factor solution is the high correlation between these factors [25,26,28]. However, recent studies have proven the imposition of a simple structure could inflate correlations between modeled factors with simulated data (e.g., [29]). Thus, as suggested by [30], “taking these correlations as justification for unidimensionality risks a self-fulfilling prophecy of simplicity begetting simplicity.” Given these controversies, the current study considers both the unidimensional and multidimensional structure of the GHQ-12.

Thus, although there are some studies that have looked at the psychiatric comorbidities of MS, much less is known about how MS is associated with other dimensions of mental health, such as social dysfunction and anhedonia and loss of confidence in a large nationally representative survey from the United Kingdom. The objective of the current study is to study how MS can affect general mental health and dimensions of mental health. The current study hypothesized that the GHQ-12 has three underlying factors labeled as GHQ-12A (social dysfunction and anhedonia; 6 items), GHQ-12B (depression and anxiety; 4 items), and GHQ-12C (loss of confidence; 2 items). Moreover, people with MS have worse general mental health and dimensions of mental health.

## 2. Methods

### 2.1. Data

This study used data from Understanding Society [31]. The current study used data in Wave 10, which was collected between 2018 and 2019 [31]. All data collections have been approved by the University of Essex Ethical Committee. Participants completed informed consent before participating in these studies. Participants with any missing variables of interest were removed from further analyses. Among them, there were 78 people with MS with a mean age of 52.19 (S.D. = 12.94) years old and 25.64% males and 38, 516 people without MS with a mean age of 49.10 (S.D. = 18.24) years old and 44.27% males who indicated that they were not clinically diagnosed with MS.

### 2.2. Measures

#### 2.2.1. MS

Participants answered the question “Has a doctor or other health professional ever told you that you have any of these conditions? Multiple sclerosis.” to indicate if they have been diagnosed with MS. Self-reported MS is a valid measure of MS status and has been used in various studies (e.g., [32,33]).

#### 2.2.2. Mental Health

Mental health was measured using the GHQ-12, which is a 12-item unidimensional measure of mental health [15]. The GHQ-12 used the Likert method of scoring ranges from 0 (“Not at all”) to 3 (“Much more than usual”). A summary score across all the 12 items was used to represent general mental health. A higher score means worse mental health. For the purpose of the factor analysis, the GHQ-12 was scored from 1 (“Not at all”) to 4 (“Much more than usual”).

#### 2.2.3. Demographic Controls

Demographic controls in the model include age (continuous), sex (male = 1 vs. female = 2), monthly income (continuous), highest educational qualification (college = 1 or below college = 2), legal marital status (single = 1 vs. married = 2), and residence (urban = 1 vs. rural = 2).

### 2.3. Analysis

#### 2.3.1. Factor Model

A confirmatory factor analysis (CFA) with oblique rotation was applied to the GHQ-12 dataset on MATLAB 2018a with native MATLAB function with a pre-specified number of factors of 3. The three factors are expected to be GHQ-12A (social dysfunction and anhedonia; 6 items), GHQ-12B (depression and anxiety; 4 items), and GHQ-12C (loss of confidence; 2 items). Both the GHQ-12 summary score and factor scores are standardized (mean = 0, std = 1).

#### 2.3.2. Linear Model

First, four generalized linear models were constructed using demographics including age, sex, monthly income, highest educational qualification, legal marital status, and residence as predictors to predict general mental health and dimensions of mental health including GHQ-12A (social dysfunction and anhedonia; 6 items), GHQ-12B (depression and anxiety; 4 items), and GHQ-12C (loss of confidence; 2 items) in healthy controls. Second, the demographics from people with MS were taken from the predicted general and mental health scores. Third, one-sample t-tests were used to compare the differences in the predicted scores and expected scores in people with MS. This approach is more advantageous than paired sample t-tests as it can control for demographic confounds and can deal with unbalanced sample size.

## 3. Results

Descriptive statistics can be found in Table 1. The CFA yielded three interpretable factors including GHQ-12A (social dysfunction and anhedonia; 6 items), GHQ-12B (depression and anxiety; 4 items), and GHQ-12C (loss of confidence; 2 items). The loadings of these items can be found in Table 2.

The current study found that there is a main effect of age (F(1, 26,387) = 184.31, *p* < 0.001), sex (F(1, 26,387) = 231.65, *p* < 0.001), monthly income (F(1, 26,387) = 48.59, *p* < 0.001), marital status (F(1, 26,387) = 103.02, *p* < 0.001), and residence (F(1, 26,387) = 24.74, *p* < 0.001) on the GHQ-12 summary score in healthy controls. However, the main effect of the highest educational qualification was not significant. Moreover, there was a significant main effect of age (F(1, 26,387) = 59.86, *p* < 0.001), sex (F(1, 26,387) = 64.31, *p* < 0.001), monthly income (F(1, 26,387) = 37.29, *p* < 0.001), highest educational qualification (F(1, 26,387) = 22.31, *p* < 0.001), marital status (F(1, 26,387) = 58.36, *p* < 0.001), and residence (F(1, 26,387) = 11.51, *p* < 0.001) on GHQ-12A in healthy controls. The current study also found a main effect of age (F(1, 26,387) = 449.68, *p* < 0.001), sex (F(1, 26,387) = 317.58, *p* < 0.001), monthly income (F(1, 26,387) = 19.32, *p* < 0.001), marital status (F(1, 26,387) = 37.73, *p* < 0.001), and residence (F(1, 26,387) = 21.84, *p* < 0.001) on GHQ-12B in healthy controls. However, the main effect of monthly income was not significant. Finally, there was a significant main effect of age (F(1, 26,387) = 482.51, *p* < 0.001), sex (F(1, 26,387) = 119.23, *p* < 0.001), monthly income (F(1, 26,387) = 131.94, *p* < 0.001), highest educational qualification (F(1, 26,387) = 19.00, *p* < 0.001), marital status (F(1, 26,387) = 181.41, *p* < 0.001), and residence (F(1, 26,387) = 17.35, *p* < 0.001) on GHQ-12C in healthy controls.

The main finding of the current study was that participants with MS have worse general health (t(77) = 2.53, *p* < 0.05, Cohen’s d = 0.34, 95% C.I. [0.07, 0.61]), more social dysfunction and anhedonia (t(77) = 3.07, *p* < 0.01, Cohen’s d = 0.43, 95% C.I. [0.15, 0.71]), and more loss of confidence (t(77) = 2.73, *p* < 0.01, Cohen’s d = 0.37, 95% C.I. [0.10, 0.64]). However, the effect of MS on depression and anxiety was not significant (Figure 1).

## 4. Discussion

The aim of the current study was to investigate how general and dimensions of mental health are affected by MS. By using factor analysis, generalized linear models, and one-sample t-tests to analyze data from 78 people with MS with a mean age of 52.19 (S.D. = 12.94) years old and 25.64% males and 38,516 people without MS with a mean age of 49.10 (S.D. = 18.24) years old and 44.27% males from Understanding Society, the current study found that there are three underlying factors of the GHQ-12, and the general mental health, social dysfunction and anhedonia, and loss of confidence are affected by MS.

In the current study, the factor analysis yielded three factors including GHQ-12A (social dysfunction and anhedonia; 6 items), GHQ-12B (depression and anxiety; 4 items), and GHQ-12C (loss of confidence; 2 items). The three-factor structure solution found in the current study is largely consistent with previous studies that identified three factors in GHQ-12 [16,22,24,25,26,27,28,34]. Moreover, as shown in Table 1, the factor loadings were high in the current study.

The finding that general mental health is affected by MS seems to be consistent with previous studies [7,8]. Poor mental health in people with MS can contribute to the development of physical and psychiatric illnesses. People with MS may experience the unpredictability of MS, sudden changes in their life, and invisible symptoms such as cognitive impairment. Moreover, they may have insatiable symptoms such as “financial complications, feeling the loss of control, and navigating unexpected decisions [35,36]”, which then leads to worse mental health.

The current study also found people with MS have more dysfunction and anhedonia and loss of confidence. However, the current study did not find that MS affects depression and anxiety, which seems to be inconsistent with the literature, which found that MS patients have more prevalence of anxiety and depression disorder as well as more anxiety and depression symptoms [1,37]. This finding could be explained by the way depression and anxiety were assessed. Specifically, in the GHQ-12, depression and anxiety were assessed by four items that ask about “loss of sleep”, “problem overcoming difficulties”, “unhappy or depressed”, and “general happiness”. However, other inventories specifically designed to assess anxiety and depression consist of many more domains that ask about symptoms of anxiety and depression. This finding may indicate that GHQ-12 may not be a good choice for measuring depression and anxiety in people with MS.

There are several pathways that could explain the association between MS and mental health. For instance, lesions in several brain regions could cause poor mental health in people with MS. Specifically, endocrine systems, lesion locations, and medications could be the contributing factors as reviewed by [8]. Second, neuroinflammation could explain the associations between MS and personality traits, with subclinical neuroinflammation predicting psychological change [38,39]. Finally, psychosocial factors including invisible symptoms may contribute to mental health in people with MS [8].

Despite the strength of the current study, there are also some limitations. First, the current study is cross-sectional, which makes it hard to establish causality in the current study. Future studies should use a longitudinal approach to establish causality if possible. Second, the current study was based on self-reported measures; future studies should use more objective assessment to enhance the contribution of the current study. Third, the current study focused on participants from the United Kingdom, which may make it hard to generalize the current findings to other countries and cultures. Multi-country studies are needed. Finally, the current study did not measure the specific stage of MS (e.g., relapsing, remitting, etc., ref. [39] and invisible symptoms of MS, which could cause bias. Future studies should control these measures.

Taken together, the current study factor analyzed the GHQ-12 and investigated how general and dimensions of mental health could be affected by MS. The current study found that general and dimensions of mental health are affected by MS except for depression and anxiety. Thus, effective mental health management in MS patients is important given that mental health in people with MS is linked to the onset of MS and disease progression/relapses [40,41,42]. Routine assessments of mental health in MS patients would be important and can be possibly achieved by the GHQ-12, in which patients can easily report their mental health within a minute. Neurologists should also work together with mental health specialists to ensure that both physical and psychological needs are met [4]. In addition, social media could be utilized to better manage mental health in MS patients. For instance, 24/7 mental health consultancy can be made available via social media. 

## Figures and Tables

**Figure 1 jcm-11-07483-f001:**
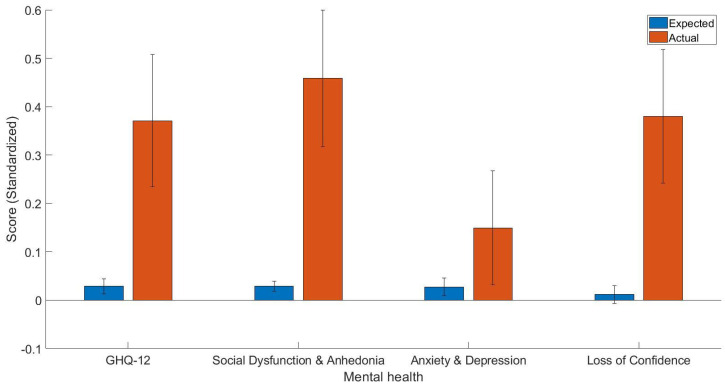
The expected and actual scores of general and dimensions of mental health (standardized).

**Table 1 jcm-11-07483-t001:** The demographic characteristics of healthy controls and MS patients.

	Healthy Controls		MS Patients	
	Mean	S.D.	Mean	S.D.
Age	49.10	18.24	52.19	12.94
Monthly income	1642.32	1502.97	1490.91	875.54
	N	%	N	%
**Sex**				
Male	11,685	44.27	20	25.64
Female	14,709	55.73	58	74.36
**Highest educational qualification**				
Below college	16,787	63.60	42	53.85
College	9607	36.40	36	46.15
**Legal marital status**				
Single	18,950	49.20	41	52.56
Married	19,566	50.80	37	47.44
**Residence**				
Urban	30,110	78.18	57	73.08
Rural	8406	21.82	21	26.92

**Table 2 jcm-11-07483-t002:** The factor loadings for the three-factor structure of the GHQ-12.

GHQ-12 Items	GHQ-12A (Social Dysfunction and Anhedonia; 6 Items)	GHQ-12B (Depression and Anxiety; 4 Items)	GHQ-12C (Loss of Confidence; 2 Items)
Concentration	**0.53**	0.25	−0.10
Loss of sleep	0.00	**0.66**	0.05
Playing a useful role	**0.67**	−0.16	0.16
Constantly under strain	**0.79**	−0.14	0.00
Problem overcoming difficulties	−0.01	**0.88**	−0.07
Unhappy or depressed	0.08	**0.56**	0.21
Losing confidence	**0.68**	0.22	−0.15
Believe worthless	**0.71**	−0.03	0.04
General happiness	0.03	**0.53**	0.34
Capable of making decisions	0.00	0.23	**0.70**
Ability to face problems	0.10	0.02	**0.74**
Enjoy day-to-day activities	**0.56**	0.10	0.10

Factor loadings for questions that load the most heavily on a certain factor were marked in bold.

## Data Availability

Publicly available datasets were analyzed in this study. This data can be found here: https://www.understandingsociety.ac.uk.

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
