# Peer review of "Understanding the Effect of Multiple Sclerosis on General and Dimensions of Mental Health"

_jcm, 2022, doi:10.3390/jcm11247483_

Round 1
Reviewer 1 Report
The objective of the paper is relevant, both from a social and scientific point of view. We agree that more rigorous studies are needed to advance the scientific knowledge of mental health in Multiple Sclerosis patients, and furthermore, it is crucial to know how this disease generates mental dysfunctions. However, the study needs a greater theoretical foundation, it does not refer so much to the methodological relevance of the GHQ-12 but to the review carried out of studies on mental health, Multiple Sclerosis disease and its dimensions.
In the Method section nothing is indicated regarding a possible gender perspective applied to the study. Nor is anything included about the different types of Multiple Sclerosis (relapsing, remitting, etc.; see, for example: Rossi, S. et al. 2017). Nor has the issue of the invisible symptoms of Multiple Sclerosis in mental health been studied. All these aspects are relevant to better understand the relationships of the phenomena studied. It would have been important to include more scales in the research instrument so that the analyzes were more relevant and robust.
The Results section requires greater detail in the presentation of the most significant results.
But, above all, the article has many shortcomings in the Discussion section. In this extremely brief section, the implications (at different levels) of the results are not detailed. It is necessary to introduce more and better interpretations about the implications of the results obtained.
Where it is stated that the current study found that MS affects mental health dimensions except depression and anxiety, this answer is not rigorously argued. Above all, it cannot be answered by this finding that the GHQ-12 is not a good choice for measuring depression and anxiety in people with MS.
The conclusions of the study are very vague and generic: It indicates that the effective management of mental health in MS patients is important since mental health in people with MS is related to the onset of MS and the progression / relapses of the disease . It also concludes that doctors and other health care providers need to devise ways to improve the mental health of people with MS. These conclusions should serve for the author to discuss and make concrete proposals of what his results imply. What do the results imply at the level of health policy, of mental health policies... What advances do these results (these findings) imply, what is the exact contribution of this research and how can it be used to introduce improvements in its scientific field.
Author Response
Dear Reviewer,
Thanks for reviewing my manuscript. Please see below for my response to your comments:
- I have added theoretical foundations regarding why MS is associated with mental health problems (see page 2).
- I have added the fact that some variables that may be of interest were not controlled in the current study as a limitation (see page 6).
- The reason why depression & anxiety is not found in the current sample was rewritten (see page 5-6).
- Implications have been further discussed (see page 6).
Reviewer 2 Report
I appreciate the research presented and the quality of the article as a whole. I made only a few minor suggestions in the sections: Introduction, Results, and Discussion.
Introduction
Point 1 - I suggest that the expression “...to avoid misunder...” (page 1, lines 35-36) could be developed (just enough to explain a little more). My suggestion refers to the following question: In what contexts is the indication of clarification directed at family members, society in general, and health services, among others? And what are the reasons, that is, what misunderstandings happen, for example? In this way, my suggestion is only complementation of the presented idea.
Comment: I congratulate the authors for explaining the use of the metric in its one-dimensional and multidimensional dimensions of the GHQ-12.
Method (Comment)
Comment: The method was presented clearly and adequately structured according to the objective of the study [to study how MS can affect general mental health and the dimensions of mental health], and its respective hypotheses [on the dimensions of GHT-12 and on the mental health conditions of people with MS – “people with MS have worse mental health in general and in the dimensions of mental health”].
Results
Point 2 – I recommend including Figure 1 in the presentation of the results. This figure is not in the analyzed file.
Discussion
Point 3 – I recommend attention to the following statement “This finding may indicate that GHQ-12 may not be a good choice for measure depression & anxiety in people with MS” (page 5, lines 182-184). Due to the very limits of the study presented by the authors.
Point 4 – I suggest the inclusion of a paragraph contributing to the study for clinical practice, management, and policymakers, given the findings that provide multidisciplinary subsidies for those interested in this debilitating human problem - Multiple Sclerosis. And in light of the statement, “Accumulating evidence has suggested that mental health could have an impact on one’s quality of life, and disease progression in MS patients.” (page 2, lines 43-44).
Author Response
Dear Reviewer,
Thanks for reviewing my manuscript. Please see below for my response to your comments:
- I have explained misunderstandings (see page 1).
- I have added Figure.1 (see page 5).
- The reason why depression & anxiety is not found in the current sample was rewritten (see page 5-6).
- Implications have been further discussed (see page 6).